# Relationship between Objectively and Subjectively Measured Physical Activity in Adolescents during and after COVID-19 Restrictions

**DOI:** 10.3390/bs11120177

**Published:** 2021-12-11

**Authors:** Armando Cocca, Klaus Greier, Clemens Drenowatz, Gerhard Ruedl

**Affiliations:** 1Department of Sport Science, University of Innsbruck, 6020 Innsbruck, Austria; nikolaus.greier@uibk.ac.at (K.G.); Gerhard.ruedl@uibk.ac.at (G.R.); 2Physical Education and Sports, Division of Physical Education, University of Education Stams–KPH-ES, 6422 Stams, Austria; 3Division of Sport, Physical Activity and Health, University of Education Upper Austria, 4020 Linz, Austria; Clemens.drenowatz@ph-ooe.at

**Keywords:** active behaviors, self-report, accelerometers, perception/reality, adolescents

## Abstract

Background: Studying the relationship between subjectively and objectively measured physical activity (PA) can provide viable information on youths’ behaviors. However, the restrictions due to COVID-19 pandemic, which reduced children’s possibilities to be active, may negatively affect it. The aim of this study was to assess the relationship between subjectively and objectively measured PA levels (light, moderate, vigorous, and moderate-to-vigorous) during COVID-19-based restrictions and after they were lifted, and to determine whether such relationships changed in these two periods. Methods: A total of 26 adolescents (58% girls; mean age = 12.4 ± 0.5) wore accelerometers during public restrictions and after they were removed. Participants also completed the International Physical Activity Questionnaire during the same periods. Results: High significant correlations were found at all levels of PA (r = 0.767–0.968) in both time periods, except for moderate PA during restrictions. Comparing the two periods, significantly higher correlations were found for moderate PA (*p* < 0.001) and moderate-to-vigorous PA (*p* = 0.003) after restrictions were lifted. Conclusions: In this highly active cohort of adolescents, results emphasize the potential threat of lockdown conditions for youths’ ability to accurately perceive their behaviors, with possible detrimental consequences on the short- and long-term health.

## 1. Introduction

Physical activity (PA) is an integral element in youths’ life. In fact, an adequate amount and type of PA contribute to children’s and adolescents’ development in different ways [1]. For instance, in the physical area, PA promotes positive metabolic function, therefore helping the body work in optimal conditions and carry out the necessary actions to maintain homeostasis and trigger overall physical development [2]. Additionally, there is a strong association of PA with physical fitness, which is composed of several elements, such as cardiorespiratory fitness, muscular strength and power, as well as flexibility, [3]. In fact, PA programs and active behaviors have been shown effective for the proper development of the individual components of physical fitness [4]. Moreover, PA contributes to increasing motor skills, coordination, balance, and other physical parameters that are essential for any individual to properly act and react in their environment, and, consequently, to be efficient [5]. Regarding the psychological sphere, the positive relation between PA and cognitive functioning and mental health has been demonstrated in previous research [5]. Also, decision-making processes and critical thinking are strongly associated with different types of PA [6]. In terms of social development, kids who engage in organized or unstructured PA have higher opportunities to achieve social integration and a more precise perception of themselves and others [7]. Furthermore, the role of PA in fostering cooperation, teamwork, and the feeling of respect towards others is well known and thoroughly investigated [7]. PA also plays a role in the short- and long-term prevention of diseases, both in the physical and social-psychological areas [8]. For instance, higher PA levels are associated with a lower risk of developing early obesity and diabetes, as well as cardiovascular issues and other nontransmissible diseases in adulthood [9]. In addition, its positive effect on mental issues, such as stress, anxiety, or depression, and on cognitive diseases, such as Alzheimer’s or Parkinson’s, have been shown in previous studies [10].

Maintaining recommended PA levels is hence essential for youth to ensure immediate benefits in their life, and to increase the chances of healthier adulthood [11]. Unfortunately, in 2016 globally 81% of adolescents aged 11–17 years did not meet such PA recommendations, i.e., 60 min of moderate-to-vigorous PA per day [12]. Insufficient PA has become even more prevalent during the COVID-19 pandemic, due to lockdowns and public restrictions [13].

The study of youths’ PA habits has been a focal approach of many scientific works in the last decades [12,13]. Scientists have used different approaches, which may be discerned based on the strategy used for assessing the amount/levels of PA: in some cases, an assessment is carried out with objective measurements, involving the employment of technological devices, such as accelerometers, GPS, or armbands [14]. Such an approach tends to deliver more precise measurements, but it requires higher financial resources to acquire a sufficient number of devices for data collection in larger samples. Objective assessments also do not provide any information on context or type of PA. In other cases, researchers utilized questionnaires, which focus on self-reported data based on one’s own perception of their daily activities, and therefore are subjective [15]. While this is an easy and fast method to obtain information from larger portions of a population, subjective perceptions may diverge from the real PA [16]. With their own limitations, both strategies are widely used in sciences, and the application of one or another may depend on contextual features. The implementation of both methods simultaneously may provide more detailed insight into people’s activity habits and their ability to properly perceive themselves. A primary aim of studies in this field is the validation of PA assessment tools. Marasso et al. [17], in their systematic review, highlighted 13 studies in the field, all of them hinting at a moderate relationship between self-reported and objective PA in children. The authors conclude that using both methods at the same time provides a more comprehensive description of youths’ PA. According to Hagstromer et al. [18], individuals tend to overestimate their PA, although the correlation with objective measures is significant. In a similar manner, previous research by Hart et al. [19] showed positive convergence of scores between accelerometry-based and self-reported PA. Beyond the purpose of validating scientific tools for the assessment of PA, the study of the relation between perception and reality has also focused on understanding potential discrepancies and pointing out associated benefits (absent or low differences) or risks (in case of both over and underestimation of the reality). For instance, accurate perception of one’s own PA is associated with higher autonomous motivation towards being active, which results in higher PA levels in youth [20]. A discrepancy between these two variables, on the other hand, may lead to worse body mass index [21]; underline potential mental issues [22]; trigger inadequate contextual responses [23], and cause lower current PA levels [20]. Additionally, a stream of research in this field has worked on the understanding of discrepancies between objective and subjective PA under particular conditions (adverse or uncommon personal or environmental circumstances): for example, Makarewicz et al. [24] found out that the relationship changes or even disappears in the presence of mild cognitive impairment; self-reported PA becomes limited, compared to objective measures, in case of physically limiting issues, such as chronic fatigue syndrome [25]; changes in the relationship between objective and subjective PA due to specific situations are further examined in other previous studies [26,27,28,29].

In line with this latter stream of research, the onset and spread of the COVID-19 disease has forced governments to take restrictive measures, pushing individuals to readapt their habits while being in a stressful situation and at the same time reducing the opportunities for children and adolescents to be active due to, for instance, closure of sports centers, social distancing, or limited possibilities to use outdoor areas [13,30]. As suggested by previous studies, such particular conditions may affect and modify the subjective perception of PA, and, consequently, its correlation with actual PA carried out. This may also depend on the fact that the more active a person is, the better is their perception of their own PA [20]: as the COVID-19 restrictions reduce youths’ chances to be active, they may lead to lower engagement in PA [31] and, consequently, less accurate PA self-perception. This represents a potential threat to youths’ willingness to maintain active habits, and to their current and future health status. As evident, many studies have focused on the relationship between objective and subjective PA in different populations; however, none has investigated how the issuing of social and environmental restrictions due to the spread of COVID-19 has modified such a relationship and, potentially, increased the risk of health issues in youth. Moreover, previous studies have either assessed such a relationship in a single time point or by comparing two conditions simultaneously (for instance, comparison of individuals with and without mental impairment with a single observation [24,25,26,27,28,29]. In this sense, our study proposes a different approach as it aims at comparing the relationship between subjectively and objectively measured PA in the same participants at two different time points and under different circumstances, i.e., during a period of COVID-19 restrictions and after these restrictions were lifted. To our knowledge, this approach is novel and may bring more precise information on concrete self-perception changes in an adverse situation. Although previous research on objective vs. subjective PA in particular circumstances showed contrasting results, we hypothesize that the limited opportunities of engaging in PA during the restriction period may have a detrimental effect; hence, while the objective-subjective relationship is expected to remain significant throughout the study period, it should be significantly stronger in the absence of restrictions.

## 2. Materials and Methods

### 2.1. Design

This study is based on an observational design. The variables studied were objective amount of weekly PA and perceived amount of weekly PA.

### 2.2. Sample

The sample was composed of 26 secondary school students (11 boys, 15 girls) from two schools in Tirol, Austria. Participants had an average age of 12.42 years, average height of 157.31 cm, and average weight of 45.96 kg.

### 2.3. Instruments

Objective Physical Activity. For the objective assessment of PA, the accelerometry devices GENEActive (Activinsights Ltd., Kimbolton, UK) were used. These accelerometers are worn on the wrist and make use of the program GENEActivPCSoftware version 3.3 (ActivInsights Ltd., Kimbolton, UK) for individual device settings and of R Studio for raw data collection. The GENEActiv has been used in research from different fields, including medicine [32], health [33] and PA [34]. The validity and reliability of these instruments in children and adolescents have been shown previously [35].

*Subjective Physical Activity.* The short form of the International Physical Activity Questionnaire (IPAQ) [36] was employed for the participants to report their PA during the previous week. This questionnaire consists of asking participants how long (specifying the number of days of the week and the average time spent each day, in hours and minutes) they are engaged in low (LPA), moderate (MPA), and vigorous PA (VPA). A brief description of the activities involved in each of the PA levels (including typical daily tasks such as gardening or housework) is provided for the items in order to make them easier to understand. The IPAQ short form includes seven questions (two on VPA, two on MPA, two on LPA, and one on weekly sedentary time (which was not considered in this study). There are different ways of calculating respondents’ scores, allowing to obtain total weekly minutes and/or total Metabolic Equivalents of Task (METs). In our study, total weekly minutes were calculated by multiplying the number of days per week of activity with the average daily time spent at each PA level. Its validity in samples of adolescents has been established in previous studies [36].

### 2.4. Procedure

After obtaining permission from school principals and parents, who signed informed consent on behalf of their children, appointments were arranged for the setup and delivery of the accelerometers, which took place at the schools, before the start of the regular school day. The devices were the property of the researchers’ institution and were only loaned to the participants during the assessment periods. Each participant was assigned one GENEActive device (recognized with a unique serial number), so they did not have to share and could not exchange their assigned device with other participants. This was particularly important as contact avoidance was one of the core restrictions during the lockdown period. A brief explanation of how the devices work, and how they should be used, was given in advance. Height and weight were measured on-site using an electronic scale (SECA^®^ 803, Hamburg, Germany) and stadiometer (SECA^®^ 217, Hamburg, Germany), and were then entered in the GENEActivSoftware, together with the personal data from each participant (such as date of birth or laterality). Once the data was entered, the recording of participants’ activity started from the moment in which they wore it. The participants were also given a form for reporting the date and duration of any unusual activity during the recorded week, for instance, in case they had to take the accelerometer off for any reason. However, they were informed that GENEActive accelerometers are water-resistant and can be worn during sleeping as well, and, therefore, they should not have been taken off during the whole assessment period. After 7 days, a second meeting was held at the schools involved. Devices were returned and participants were asked to fill in the IPAQ questionnaire to report their PA in the previous 7 days. The same procedure was repeated twice, the first time during the period of restrictions (beginning of May 2021) due to the spread of COVID-19, and the second when no restrictions were in place anymore (middle of June 2021).

### 2.5. Data Analysis

Raw data from the accelerometers, based on EPOC lengths of 60 s, was collected using the GENEActivSoftware, and successively processed using a macro from R Studio. The macro delivers the duration of PA at different levels (vigorous, moderate, and/light), split in each of the 7 days recorded. The total time spent at the different PA levels during the whole week was obtained summing up the durations for each of the individual days. Regarding the IPAQ, total week time spent at the different PA levels was calculated directly from the raw data, by multiplying the reported days spent at each level and the average daily duration of the activity at the same level. This calculation was executed after carrying out the data control and depuration as established in the official IPAQ manual. For both objective and subjective assessments, the variable Moderate-to-vigorous PA (MVPA) was calculated by adding up the time spent on vigorous and moderate PA.

After checking for data normality by means of the Saphiro-Wilk test, over/underestimations were calculated by subtracting accelerometer data to IPAQ scores for each level of PA. Positive results of the subtraction were classified as overestimations (IPAQ score > Accelerometer scores); negative results were classified as underestimations (Accelerometer scores > IPAQ scores). Paired comparisons were carried out employing paired-sample *t* tests (VPA during restrictions, MVPA during and after restrictions) and Wilcoxon signed signed-rank tests (LPA and MPA during and after restrictions, VPA after restrictions) to assess the significance of the resulting over/underestimations. Correlations between subjective and objective PA at the two-time points were examined using Pearson’s (VPA during restrictions, MVPA during and after restrictions) and Spearman (LPA and MPA during and after restrictions, VPA after restrictions) method, based on the data distribution in each pair of variables. Successively, Fisher Z transformations were performed in order to compare the values from the two-time points. Fisher Z transformations are considered a valid method to assess differences between correlations for both Pearson’s and Spearman methods [37].

## 3. Results

Descriptive results for the sample are shown in Table 1 below.

Both during restrictions and after them, according to accelerometry, participants achieved the minimum recommended weekly MVPA of 60 min per day (average MVPA during restrictions = 72.52 min/day; average MVPA after restrictions = 89.32 min/day).

For all variables, subjective PA was overestimated compared to the objective one, with the exception of MPA during lockdown, when 65.4% of the participants underestimated their total minutes per week at a moderate intensity. Significant differences were found for the overestimated subjective perception of VPA and MVPA both during and after restrictions (Figure 1).

Correlations among objective and subjective PA during restrictions and after restrictions were lifted and showed strong significant outcomes for all variables, except for MPA during restrictions (r = 0.381, *p* = 0.055). Descriptive data of each variable studied and detailed results from the correlation analyses can be found in Table 2.

In most variables, the adolescents in our study overestimated their actual PA time. This is not true only for MPA during restrictions, as participants reported lower MPA time than objectively measured (subjective MPA = 244.23; objective MPA = 268.85). Fisher Z comparisons between correlations during and after restrictions showed no significant differences in most variables. However, statistical significance was found for MPA (*p* < 0.001) and for MVPA (*p* = 0.003). In both cases, higher correlations were found in the period with no restrictions compared to that when restrictions were in force (Table 2).

## 4. Discussion

The objectives of this study were to assess the correlation between objective and subjective PA levels in youth in two different situations: once, with COVID-19-based restrictions in force, limiting individuals’ possibilities to move in their community and limiting their use of facilities normally employed in their daily living; and a second time immediately after the restrictions were lifted, allowing the adolescents to return to their normal life and daily routines.

Generally, the studied population was highly active as, based on population average, the WHO recommendations of daily 60 min of MVPA were clearly exceeded both during and after the COVID restrictions. In addition, active behaviors at different participants’ PA levels were clearly higher after restrictions were lifted (Table 2).

As expected, participants tended to significantly overestimate their actual PA at the MVPA intensity. This is in line with the body of literature on the topic, independently of the type of participants and the personal/environmental conditions during the study [18,38,39,40].

However, as literature shows, correlations are considered the reference indicator of the relationship between objectively and subjectively measured PA. In this sense, our findings show a highly positive perception of the young participants regarding their active behaviors. Regardless of the different levels of PA, these individuals were able to effectively address their actual engagement in PA during a regular week, this playing a fundamental role in allowing them to maintain higher PA levels over time [20]. Perhaps, this may be due to a better ability to reflect on abstract concepts that accompanies this phase of youths’ life [41]. When perception of PA, and real PA, are related, indicating a proper notion of oneself and one’s interaction with the environment, there is a higher chance that an individual engages in PA and is more motivated towards being active [20,41]. However, it should be addressed that this correlation could not be confirmed for MPA during the period with restrictions. Nonetheless, once the public restrictions were lifted and the participants were allowed to engage again in their normal activities, with all sports facilities open and accessible, the correlation became significant. Firstly, this may be associated with increased barriers towards being active during COVID-19 lockdowns or restrictions. In such conditions, youth have fewer chances to be active and their PA levels and amount may drop [42]. It has been suggested that the more active the children are, the more able they are to accurately perceive their PA behaviors [20]. Therefore, our findings for MPA may be justified by the reduced time engaged in PA during the restrictions, which is also reflected by the total amount of activity during and after restrictions measured via accelerometry. A second potential justification of our outcomes regarding MPA may be found in the impact of mood and psychological status on perceived PA. In fact, some authors have pointed out a negative relationship between perceived anxiety and mood associated with COVID-19 confinement, and self-reported PA [43,44]. Lower mental well-being determined by the restrictions and, consequent to them, the reduced possibilities to be active outdoors and socialize through sports with others, seem to have a significant impact on perceived PA [45]. Also, it may be pointed out that MPA is the only PA that our participants underestimated during the restrictions. This is in line with Canning et al. [46], who also reported a tendency in underestimating MPA in their study. Also, Szymczak et al. [47] suggest that MPA may not be associated with better individual self-perception compared to other PA intensities. Hence, we could assume that both these factors may have played a role also in our sample’s inability to accurately perceive their MPA. Furthermore, the fact that our sample was highly active and, especially when restrictions were removed, exceeded the recommended daily PA, for youth may have contributed to the differences in MPA correlations between the two periods: in fact, as previously pointed out, it seems that higher PA leads to more accurate perception. Finally, our findings on MPA may help explain the differences found in MVPA correlations. Although both during restrictions and after they were lifted MVPA correlations remained significantly positive, their comparison addressed a statistical difference, with subjective and objective MVPA being better correlated with no COVID-19 restrictions in force. However, as mentioned above, and based on the fact that this variable was calculated as the aggregate of MPA and VPA scores, such difference is likely to depend entirely on the significant changes in perceived MPA between the two periods.

## 5. Conclusions

This research proposed a novel approach to the study of the relationship between objective and subjective PA in youth. Different than the existing body of literature, mostly based on single observations at one-time point, our main approach was to track a sample of adolescents during the period of COVID-19-based restrictions and after these restrictions were lifted, in order to examine changes in such a relationship between the two conditions. The constraints and limitation in adolescents’ freedom during the pandemic have negatively affected particularly their ability to perceive the amount of MPA they actually carry out. This is also reflected in a lower perception of MVPA, intensity commonly associated with health benefits. Although VPA is considered the most impactful intensity for health, it requires high effort and, consequently, cannot be maintained for long intervals. On the contrary, MPA represents a larger portion of youth’ daily and weekly PA, and therefore, we must consider the potential risks associated with a worse perceptiveness of it. In the face of potential future situations similar to those of the COVID-19 pandemic, efforts should be put in delivering alternative strategies for engaging in moderate-to-vigorous physical activity from home, or with limited space and equipment. In this sense, the potential of social media, virtual rooms, video games, and remote communication should be explored in a deeper manner so to develop exercise programs tailored to adolescents’ needs.

### Limitations and Considerations

The main limitation of this study is the sample size. Due to the methodology of our research (intending to track the same participants throughout the study period) and the limited possibilities of contact with schools and individuals during the lockdown period, it was not possible to recruit a larger number of participants. Therefore, our results should be taken with caution and considered as a starting point for further research taking a similar approach. However, as previously stated, the chosen methodology is a novelty compared to the existing literature in the field, as, to the best of our knowledge, no study to date has attempted to assess changes in the objective-subjective PA relationship in the same group of participants at different time points (i.e., different environmental conditions). Therefore, we decided to maintain this novel aspect despite the limitation in the size of the sample. The decision was supported by previous published works in the field that, similar to ours, have included a reduced number of participants [19,40,48].

## Figures and Tables

**Figure 1 behavsci-11-00177-f001:**
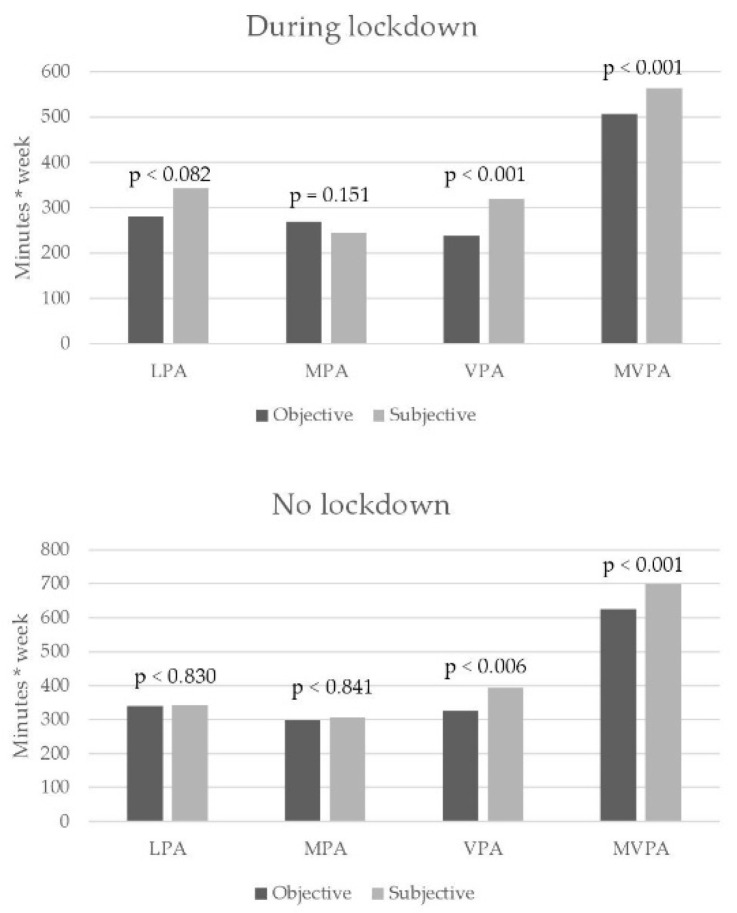
Comparison of objective and subjective PA scores during restrictions and after they were lifted. LPA = Low physical activity; MPA = moderate physical activity; VPA = vigorous physical activity; MVPA = moderate-to-vigorous physical activity.

**Table 1 behavsci-11-00177-t001:** Descriptive data of the sample.

Sample (n)	Age	Height (cm)	Weight (kg)	BMI	BMIpct (%)
Boys (11)	12.82 ± 0.41	160.12 ± 7.49	50.04 ± 7.40	19.46 ± 2.19	47.73 ± 26.40
Girls (15)	12.13 ± 0.35	155.25 ± 7.35	45.96 ± 10.05	18.94 ± 3.24	40.33 ± 30.35
Total (26)	12.42 ± 0.50	157.31 ± 7.66	47.68 ± 9.09	19.16 ± 2.80	43.46 ± 28.43

Note. **BMI** = Body Mass Index; **BMIpct** = Body Mass Index Percentile.

**Table 2 behavsci-11-00177-t002:** Analysis of correlations (**r**) between objective (via accelerometry) and subjective (via questionnaire) assessment of physical activity (PA) during the period of restrictions and after it.

Variable	P	Objective	Subjective	r	*p*
		(min)	(min)		
Light PA	R	281.73 ± 205.71	342.69 ± 289.99	0.877	<0.001
	NR	340.00 ± 196.29	343.13 ± 279.70	0.897	<0.001
Moderate PA	R	268.85 ± 110.53	244.23 ± 180.53	0.381	0.055
	NR	298.54 ± 96.30	306.67 ± 160.58	0.938	<0.001
Vigorous PA	R	238.85 ± 149.58	320.00 ± 226.01	0.948	<0.001
	NR	326.67 ± 162.35	393.33 ± 256.22	0.968	<0.001
MVPA	R	507.69 ± 197.26	564.23 ± 320.45	0.767	<0.001
	NR	625.21 ± 202.61	700.00 ± 304.97	0.949	<0.001

Note: **P** = period; **min** = minutes; R = during restrictions; NR = no restrictions; MVPA = moderate-to-vigorous PA.

## Data Availability

The data presented in this study are available on request from the corresponding author. The data are not publicly available due to privacy reasons.

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
