# Peer review of "Relationship between Objectively and Subjectively Measured Physical Activity in Adolescents during and after COVID-19 Restrictions"

_behavsci, 2021, doi:10.3390/bs11120177_

Round 1
Reviewer 1 Report
This is an interesting but not very original article. The authors compare the objective (accelerometers) and subjective (IPAQ) measurement of the physical activity of young people during the loockdown in Austria.
I have nothing to say about the methodology (even if the number of participants seems to me a little limited to reach the conclusions indicated) and the structure of the article, but rather about the originality and the theoretical references.
It would be interesting for the authors to refer to other similar research. There have been countless studies on the IPAQ and similar questionnaires (GPAQ etc.), on similar populations, and these studies should be mentioned in the introduction, also to put forward the partial originality of your work in relation to these previous studies They should also be mentioned in the conclusion too: the lockdown variable, what does it bring in addition to previous research?
Furthermore, in the introduction the authors could also indicate possible working hypotheses: what do they expect to find according to the mentioned researches?
Reviewer 2 Report
- The first question: what was the reason why the authors chose such a small number of subjects for this research? is there any logical reasoning for this number? I think the number of participants is very small and I don't know how relevant are their results.It is very important that the authors justify very clearly this very important aspect of the research undertaken....
- The citation system used in the paper is not the one required by the journal, please read the instructions for the authors
- Line 114 - Activinsights Ltd., Kimbolton, UK - did all participants afford this device or was it borrowed from one subject to another? it is not understood in the methodology how the accelerometry was used for all the people involved in the study.
- IPAQ - please add additional information about this assessment tool, how many questions it has, what was the score, etc.
- Must be specified in the methodology when the participants took down the accelerometers, when they sleep? when did they wash? please specify exactly the required information
- Point after each title of tables/figures
- Besides the correlations I think that other statistical analyzes are necessary to calculate, there are too few calculations presented
- I did not understand if the authors presented what were the limits of this study
- Institutional Review Board Statement - what was the ethical number?
- Please put abbreviation for all journals
Round 2
Reviewer 2 Report
Congratulations to the authors for the changes made, I recommend the publication